# On closing the inopportune gap with consistency transformation and iterative refinement

**Mario João Jr.**[1,2]*, **Alexandre C. Sena**[3], **Vinod E. F. Rebello**[2]

**1** Medical Sciences College, State University of Rio de Janeiro, Rio de Janeiro, Rio de Janeiro, Brazil, **2** Institute of Computing, Fluminense Federal University, Niterói, Rio de Janeiro, Brazil, **3** Institute of Mathematics and Statistics, State University of Rio de Janeiro, Rio de Janeiro, Rio de Janeiro, Brazil

* junior@lampada.uerj.br

**Data Availability Statement:** All Developer Scores for all instances are available at URL: https://doi.org/10.6084/m9.figshare.23518842.v2.

**Funding:** This work has received financial support from the Brazilian funding agency CNPq (Conselho

## Abstract

The problem of aligning multiple biological sequences has fascinated scientists for a long time. Over the last four decades, tens of heuristic-based Multiple Sequence Alignment (MSA) tools have been proposed, the vast majority being built on the concept of Progressive Alignment. It is known, however, that this approach suffers from an inherent drawback regarding the inadvertent insertion of gaps when aligning sequences. Two well-known corrective solutions have frequently been adopted to help mitigate this: Consistency Transformation and Iterative Refinement. This paper takes a tool-independent technique-oriented look at the alignment quality benefits of these two strategies using problem instances from the HOMSTRAD and BAliBASE benchmarks. Eighty MSA aligners have been used to compare 4 classes of heuristics: Progressive Alignments, Iterative Alignments, Consistency-based Alignments, and Consistency-based Progressive Alignments with Iterative Refinement. Statistically, while both Consistency-based classes are better for alignments with low similarity, for sequences with higher similarity, the differences between the classes are less clear. Iterative Refinement has its own drawbacks resulting in there being statistically little advantage for Progressive Aligners to adopt this technique either with Consistency Transformation or without. Nevertheless, all 4 classes are capable of bettering each other, depending on the instance problem. This further motivates the development of MSA frameworks, such as the one being developed for this research, which simultaneously contemplate multiple classes and techniques in their attempt to uncover better solutions.

## 1 Introduction

The process of aligning three or more biological sequences (typically protein amino acids, DNA or RNA base pairs) to identify the similarities or disparities that reflect the evolutionary, functional, or structural relationship [1] between them is generally referred to as *Multiple Sequence Alignment* (MSA). MSA is an essential step to solve problems in different areas of molecular and computational biology such as evolutionary-based studies [2], domain analysis

Nacional de Desenvolvimento Científico e Tecnológico - https://www.gov.br/cnpq) through the project Universal, process number 404087/2021-3. The funders had no role in study design, data collection and analysis, decision to publish, or preparation of the manuscript.

**Competing interests:** The authors have declared that no competing interests exist.

[2], co-evolutionary analyses [3], phylogeny inference [4], and the prediction of protein functions [5] or their three-dimensional structures [6], among others.

Note that this problem is computationally different, for example, from the process to find homologous sequences, which consists of obtaining multiple *pairwise sequence alignments* of protein or RNA sequences generally against a single reference sequence [7]. Multiple sequence alignments tend to provide more information than pairwise alignments by highlighting conserved regions of aligned residues that may be of structural and functional relevance. The focus of this paper is thus on the former, the traditional MSA problem.

Given the average length $l$ of the biological sequences being aligned, the optimal alignment of two sequences has an $O(l^2)$ polynomial complexity [8] while aligning $n$ such sequences optimally is an NP-complete problem with complexity $O(l^n)$ [9, 10]. This means that obtaining even approximate MSA solutions for scientifically relevant sequence lengths in acceptable time frames (e.g., aligning dozens of sequences with lengths of more than a few hundred residues [11]) has long motivated the development of heuristics that adopt strategies based primarily on pairwise sequence alignment.

Gaps are the spaces that are required when trying to align DNA or protein sequences so that homologous residues are matched. Over evolutionary time, a sequence may lose some residues (referred to as deletion) or have some extra residues inserted (insertion). When aligning with the original sequence, the newer sequence would require the insertion of characters to represent gaps in the locations where deletions occurred, while the original sequence would require gaps where the newer sequence has extra residues. When aligning multiple sequences, difficulties arise with regard to the positioning of these gaps. Given the complexity of determining whether the resulting gaps are caused by insertions or deletions, traditional heuristics typically do not often make this determination and instead best align sequences using gaps in positions that obtain the highest overall score. While Needleman and Wunsch first described a dynamic programming algorithm for the pairwise alignment of biological sequences [8], Gotoh extended this approach with affine gap scores [12] to address the assumption that a single large insertion/deletion (indel) event is biologically more likely to occur compared to many smaller ones with the same combined total length.

Numerous multiple sequence alignment tools have since been proposed over the last four decades [11], some of which have continued to evolve over the years [13, 14]. Interestingly, the majority of contemporary heuristic-based tools are still based on Progressive algorithms that aggregate multiple pairwise alignments. This Progressive method is considered to be one of the first practical MSA heuristics [15] and to date has maintained its three-stage workflow: In the first stage, the $n$ sequences are compared with each other, in pairs, generating a triangular matrix with the result of each comparison represented by a *Similarity Score*; The next stage creates a binary *Guide Tree* that identifies the order in which the sequences should be aligned in the subsequent stage. The Guide Tree is generated using a hierarchical clustering algorithm, such as UPGMA [16] or Neighbor-Joining (NJ) [17]. During the final Profile Alignment stage, the Guide Tree is traversed recursively bottom-up, generating a profile [18], for each non-leaf node, by aligning the two sets of previously aligned sequences from the node's children, until the complete alignment is produced.

While highly effective, one of the characteristics of solutions produced by the Progressive method can be summed up in the words of its authors: "once a gap, always a gap" [15]. In the third stage, if a gap is inserted during the creation of a profile to optimize the pairwise alignment, this gap will be propagated to all subsequent profiles generated from this node, including the final alignment, which could hinder the method's ability to find the optimal alignment [11].

The quality of an MSA heuristic is generally evaluated by comparing the accuracy of the alignments produced for a set of benchmark MSA problem instances against their corresponding reference alignments, hand-crafted by scientists, which are believed to be the most biologically appropriate for the set of sequences. One key factor that significantly influences the accuracy of progressive alignments is the degree of similarity between the sequences being aligned [19]. The more dissimilar the sequences are, the harder it appears for MSA heuristics to achieve high accuracy. The term *Identity* is often used to refer to the degree of similarity between sequences as a percentage of the positions in the pairwise alignment that have matching residues. In particular, according to Pei in [20], progressive aligners may not produce satisfactory results when Identity values fall inside the twilight zone [21], i.e., when the similarity between sequences is less than 30%.

To mitigate the problem of these inopportune gaps and improve the alignments produced by Progressive methods, researchers widely employ two other classes of heuristics based respectively on the techniques *Iterative Refinement* [22] and *Consistency Transformation* [23]. Iterative refinement-based alignments rely on post-processing an MSA solution to revise the order in which the profiles were aligned [24]. A heuristic process starts with an initial MSA, typically provided by a progressive alignment, and reaches a new one through successive iterative refinements that try to remove these gaps. These refinements are applied until a given objective function cannot be improved or a user-defined limit of attempts has been made. Alternatively, Consistency-based heuristics try to improve the profile alignment stage of the Progressive method using additional information, extracted from the sequences at an earlier stage, to reinforce the alignment of specific regions and avoid the inadvertent insertion of gaps [23]. This technique is also particularly effective for twilight zone alignments [19].

Consequently, the majority of MSA tools currently employ progressive aligners with either Iterative Refinement or Consistency Transformation. Which tool to choose and which parameters and techniques to configure are not easy decisions for the inexperienced [2, 25]. This choice becomes even harder given that alignment quality will be influenced by the quantity, lengths, and, in particular, the similarity of the biological sequences being aligned [1]. In this context, this work aims to examine the relative benefits of using combinations of different techniques in each stage of the Progressive method and to identify novel or perhaps overlooked combinations that could help scientists find higher-quality alignments.

Different from previous research that typically compares specific MSA tools [2, 23, 25, 26], this paper focuses on the common structures of such tools and the impact alternative techniques, methods, or heuristics can have on the quality of the alignments produced. The framework used in [19], which combines distinct techniques from existing tools including a variety of scoring techniques, Guide Tree clustering algorithms, and Consistency Transformation, has been extended to incorporate Iterative Refinement. This paper aims to investigate whether progressive alignments can effectively benefit from the two classic techniques used to address the issue of inappropriate gaps in the final alignment.

A thorough evaluation using protein sequences from the BAliBASE (Benchmark Alignment dataBASE) [27] and HOMSTRAD (HOMologous STRucture Alignment Database) [28] benchmarks is presented with results showing that, overall, consistency-based progressive alignments have greater accuracies than those found by both iterative refinement heuristics and even a novel combination of the two approaches. Even though the Wilcoxon signed-rank tests [29] show that the results have statistical significance, when analyzing the alignments for each problem instance on a case-by-case basis, one of the other 3 classes of heuristic (Progressive Alignment alone, Progressive Alignment with Iterative Refinement, and Consistency-based Progressive Alignment with Iterative Refinement) frequently produce a solution with better accuracy, indicating these approaches should not perhaps be ignored or discarded.

The rest of this paper is divided into four sections. The following one covers some background information on the MSA problem, while Section 3 briefly describes our MSA framework [19]. Section 4 presents a comparison of the alignment qualities of 80 different Progressive Alignment-based heuristics, while the final section closes the paper with some conclusions.

## 2 Background

The *modus operandi* of the class of Iterative Alignment tools [12, 24] is one approach to improve the results of a Progressive Alignment. These tools are generally composed of the following steps: (a) an initial multiple sequence alignment is generated using Progressive Alignment; (b) this alignment is then split into two smaller profiles (i.e., two sets of two or more aligned sequences) so that any entire column in either profile containing only gaps can be removed; (c) The two profiles are then realigned, generally using the technique employed in the earlier Profile Alignment stage, and; (d) if the resulting MSA obtains a better evaluation score, this becomes the MSA to be improved. Steps (b) to (d) are then repeated until a specified number of iterations have been completed or when no further improvements to the evaluation score are obtained.

While the method for dividing an alignment into two profiles in step (b) varies according to each tool, they all require a given criterion to be adopted to determine the allocation of each sequence to one of the respective profiles. The criteria most commonly used range from being simple (e.g., a random choice [30]), to something more elaborate (e.g., Tree Restricted Partitioning [31] presented in Section 3.5).

An important aspect of Iterative Alignments is how to evaluate the MSA at each iteration. The most widely used functions are the sum-of-pairs (SP) [9] and, its extension, the weighted sum-of-pairs (WSP) [32]. The WSP of an alignment $A$ of $n$ sequences and length $L$ is defined by:

$$WSP(A) = \sum_{1 \leq l \leq L} \sum_{1 \leq i < j \leq n} w_{i,j} \times S(a_{i,l}, a_{j,l}) \tag{1}$$

where $a_k$, for $1 \leq k \leq n$, is the $k$th aligned sequence, $a_{k,l}$ is the residue of the aligned sequence $a_k$ at column $l$, $w_{i,j}$ is the weight associated with the sequence pair $a_i$, $a_j$ ($w_{i,j} = 1$ for SP), and $S$ ($x$, $y$) is the residue substitution score of $x$ by $y$. The residue substitution matrix $S$ encapsulates the rates at which each type of residue would be substituted by other residues over time. One of the most widely adopted examples for $S$ is Blocks of Amino Acid Substitution (BLOSUM) [33].

Two of the best known Iterative Alignment tools are MAFFT [34] and MUSCLE [22]. Both follow the steps described earlier, but each of them has its own particularities. MUSCLE generates an initial MSA by passing through the progressive method stages twice and then iteratively attempts to improve it. First, it employs *k-mer* distance [35] to determine the first similarity score matrix which is used by UPGMA [16] to generate the Guide Tree and, after that find an initial MSA. With this first MSA, MUSCLE generates a second similarity score matrix using Kimura distance [36]. With the second similarity score matrix, MUSCLE again uses UPGMA to create a second Guide Tree, and then the MSA is iteratively improved. MUSCLE's iteration process uses Tree Restricted Partitioning [31] to determine the two profiles to be aligned employing WSP [32] as the objective function.

MAFFT version 5 [37] has five options that use iterative refinement: NW-NS-i, FFT-NS-i, F-NS-i, G-NS-i, and H-NS-i. The fundamental difference between these variations is the technique used in the first stage to calculate the similarity scores. While NW-NS-i and FFT-NS-i

both use the number of 6-tuples of residues shared by the two sequences as the score, the latter first transforms the sequence representation using a Fast Fourier Transform (FFT) before the calculation [34]. G-NS-i also uses an FFT, but a global alignment algorithm, like Needleman-Wunsch [8], to generate similarity scores. H-NS-i and F-NS-i use variations of the alignment tool FASTA [38]. In Stage 2, MAFFT also uses UPGMA and the objective function WSP to generate the Guide Tree, which is then optimized iteratively using Tree Restricted Partitioning in the following stage. The principal difference in relation to MUSCLE is related to how the similarity scores are calculated.

An alternative approach to address the drawback of the Progressive method is through the use of Consistency Transformation. Consistency-based Alignment assumes that multiple sequence alignments are consistent, i.e., if three sequences $a_a$, $a_b$, and $a_c$ are aligned, and residue $a_{a,i}$ is aligned with the residue $a_{b,j}$ and residue $a_{b,j}$ is aligned with $a_{c,k}$, then $a_{a,i}$ should be aligned with $a_{c,k}$. Thus, applying this transitive principle in reverse, when aligning two sequences $a_a$ and $a_c$, consistency-based MSA methods look for evidence of previous pairwise alignments involving $a_a$ and $a_c$ with other sequences (e.g., $a_b$) to reinforce alignments of specific residue pairs. Gotoh [39] refers to these consistent alignment regions as anchor points and has shown how they can be used to support multiple alignments. Two of the most well-known consistency-based alignment tools are T-COFFEE [23] and ProbCons [30].

## 3 Towards a common framework for MSA

While Section 2 cited a few examples, there has been a plethora of tools proposed that fall into one of the three aforementioned heuristic classes. Some even happen to combine characteristics of more than one class (e.g., iterative and consistency [30]). As yet there does not seem to be a single MSA tool that always produces the best alignment. Thus, the appropriate choice is left to the user's experience. Given the different classes and implementations within each class, it is often necessary to employ various tools or one with multiple configuration options and compare their alignment results manually [1]. This, however, requires various MSA processing steps to be unnecessarily repeated. For example, using Progressive Alignment, a scientist might want to compare the alignments generated by two different techniques for constructing the Guide Tree. In this case, the information necessary to calculate the similarity scores might be the same, but would be calculated twice, once for each technique.

To provide an environment where the user can efficiently evaluate different combinations of techniques and classes of MSA heuristics without having to learn to execute a suite of existing MSA tools, the work in [19] presented an initial version of an integrated framework of common MSA techniques to overcome these difficulties more efficiently. Furthermore, the framework is able to evaluate novel combinations of techniques, many of which have apparently not been investigated previously, in a coordinated manner and on a level playing field. Techniques from different state-of-the-art MSA tools have been integrated into the framework. The Clustal W [26] source code is used as the base implementation for Progressive Alignments, while Consistency Transformation implemented by T-COFFEE [23] has also been incorporated. For this paper, an iterative strategy has been added to the framework based on MUSCLE's [22] Tree Restricted Partitioning [31].

Instead of structuring the framework into three different classes, existing approaches are mapped to a unified architecture composed of 5 stages, with each having a number of alternative implementation options. An overview of the framework stages as well as the techniques currently available in each are described in the following subsections. The final subsection presents the nomenclature adopted in the following section to identify the variations and the heuristic classes that can be designed by the current version of this common framework.

## 3.1 Stage 1: Similarity scoring

The first step of any progressive alignment is to determine the degree of similarity between pairs of sequences and calculate each pair's *Similarity Score*. In general, these scores are held in a *Similarity Matrix*: a triangular matrix of dimension $n$, where $n$ is the number of sequences to be aligned. Each element $M_{i,j}$, with $i < j$, of this matrix is the similarity score between sequence $i$ and sequence $j$, calculated in this stage. In addition to the two techniques provided by Clustal W [26], known as FULL (and referred to as F in the tables of Subsection 4.3) and QUICK (Q), another three forms of calculating similarity scores, LCS (L), KMERS (K), and PROBA (P), have also been implemented. Their differences are summarized as follows:

- **FULL**: The similarity score is the Identity percentage of the alignment obtained using the Myers and Miller global alignment algorithm [40];

- **QUICK** calculates the scores using a fast approximation method [41] that simply counts the number of tuples that match between the two sequences;

- **LCS** defines the scores to be the length of the Longest Common Subsequence divided by the length of the smallest of the two sequences. LCS is generated using an optimized implementation [42] of Hirschberg's Algorithm [43];

- **KMERS** calculate the score by dividing the number of k-mers (the term $k$-mer refers to all of the subsequences of length $k$) that the two sequences have in common by the length of the smaller sequence, as used by MUSCLE;

- **PROBA** is an alternative approach used by ProbCons [30] that searches for alignments with maximum expected accuracy, using pair-hidden Markov model-derived posterior probability matrices that contain the probability of a pair of residues from the two sequences being aligned.

Note that each scoring mechanism inherently tends to incorporate a chosen scheme to "align" each pair of sequences. The framework thus opts to consider each unique alignment-scoring combination as a separate technique.

## 3.2 Stage 2: Guide tree construction

In addition to the two most commonly used hierarchical clustering algorithms to generate Guide Trees, UPGMA [16] and Neighbor-Joining (NJ) [17], both already available in Clustal W [26], two other variations of SLINK [44], called SLMIN and SLMAX, have been implemented in the framework. One motivation to include these two techniques was the fact that the authors in [45] claim that SLMIN is the best choice.

With the degree of similarity between pairs of sequences being held in a similarity matrix, the agglomerative UPGMA begins to construct its guide tree by representing each sequence as a single node and repeatedly joining the two nodes of the pair of sequences with the highest score in the similarity matrix to form a rooted tree. The similarity matrix is then altered to reflect the grouping of the two nodes into one. The two lines and columns of the similarity matrix, representing the two grouped nodes are substituted by a single line and a single column for the new root, with the scores between the remaining nodes being calculated as the average of the two scores being removed. The algorithm continues until all the nodes have been combined into a single cluster, representing the guide tree. SLMIN and SLMAX follow the same algorithm, however, the new scores are no longer the average of the two scores but rather the minimum (in the case of SLMAX) or the maximum (for SLMIN) score between pairs.

Similarly, Neighbor-Joining (NJ) is also a bottom-up agglomerative hierarchical clustering approach that uses a similarity matrix. Starting with a central node connected to all other nodes (sequences), the two least distant nodes (most similar sequences) are grouped, forming a tree that is connected to the central node. The distances between groups are calculated based on the path from the central node to each group, and the matrix is updated to incorporate the new group. This process is repeated until the completion of the guide tree.

### 3.3 Stage 3: Consistency transformation

Consistency Transformation was added to the framework through the concept of *constraint lists* as implemented in the widely adopted tool T-COFFEE [23] which uses posterior probabilities used earlier by the ProbCons [30] tool. The constraint list is used in the profile alignment stage to enforce the alignment of certain residues (anchor points) by adding constraint list values to the substitution matrix used by Clustal W.

### 3.4 Stage 4: Profile alignment

The technique used to align profiles in our framework is almost identical to the one adopted by Clustal W [26]. Default features like sequence weighting, position-specific gap penalties, and adaptive weight matrix choices were maintained and, to implement Consistency Transformation, support for the integration with constraint lists has been incorporated. However, the technique for delayed alignments, which overrides the Guide Tree order, has been disabled in order to be able to identify the impact of techniques chosen in earlier stages [19].

### 3.5 Stage 5: Iterative refinement

Each iteration of the Tree Restricted Partitioning [31] refinement algorithm is composed of applying the following steps to each edge of the Guide Tree, starting with the edges farthest from the root, and moving up towards the root: cut an edge of the Guide Tree, forming two sub-trees; considering the set of aligned sequences in a profile of each sub-tree, remove any column made up entirely of gaps; realign the two resulting profiles to create an intermediate MSA; calculate the WSP score of this intermediate MSA, if this calculated score is higher than the WSP of the current best MSA, the intermediate MSA becomes the new current best MSA solution. The iterative refinement process ends when an iteration fails to find an MSA that improves the WSP or when a certain number of iterations, defined by the user, has been reached.

While ProbCons [30] adopts a simpler approach where, during a user-defined number of iterations, sequences are randomly divided into two profiles, the MUSCLE implementation is preferred to be able to investigate the scope of improvement that can be achieved by this technique. It is important to emphasize that Iterative Refinement seeks to improve the WSP score of the generated alignment, which does not necessarily imply improving the quality of this alignment.

### 3.6 Framework variations and strategies

Currently, our framework has 20 variations of progressive alignment, given that the first stage has a choice of 5 techniques to calculate the similarity score (Section 3.1) and 4 guide tree generation algorithms in the second stage (Section 3.2). The names of the combinations, referenced extensively in Section 4.3, are illustrated in Table 1.

The following section refers to the combination of a variation (defined by Stage 1 and 2) and a heuristic class to possibly improve progressive alignment as a *strategy*. Each of the four

**Table 1. The 20 progressive alignment variations currently in the common framework.**

|  | FULL | KMERS | LCS | PROBA | QUICK |
|---|---|---|---|---|---|
| **NJ** | F.NJ | K.NJ | L.NJ | P.NJ | Q.NJ |
| **SLMAX** | F.SLMAX | K.SLMAX | L.SLMAX | P.SLMAX | Q.SLMAX |
| **SLMIN** | F.SLMIN | K.SLMIN | L.SLMIN | P.SLMIN | Q.SLMIN |
| **UPGMA** | F.UPGMA | K.UPGMA | L.UPGMA | P.UPGMA | Q.UPGMA |

MSA heuristic classes are identified by a different combination of the remaining three stages of the framework: Progressive Alignments only employs Stage 4; Consistency-based heuristics that apply an additional Consistency Transformation step (Section 3.3) before the final alignment requires Stages 3 and 4; Iterative heuristics append an iterative refinement step (Section 3.5) in Stage 5 after the profile alignment of Stage 4; finally, the combination of all three stages (Stages 3, 4 and 5) produces the somewhat less well-known class of Consistency-based Progressive Alignments with Iterative Refinement. Given the variations presented in Table 1, and these four classes, Section 4 evaluates a total of 80 different MSA strategies.

## 4 Experimental analysis

Instead of treating MSA tools as black boxes and comparing the quality of the alignments they produce, as has already been extensively covered in the literature, the aim of this section is to evaluate design alternatives combining different classical techniques or methods. For example, do Progressive Alignments benefit the best from the use of Iterative Refinement, Consistency Transformation, or even a combination of both? The quality of the alignments produced depends very much on the characteristics of the sequences themselves as well as the combinations of techniques adopted by an MSA tool, thus it is common to find a given strategy can generate better, similar, or worse alignments than other approaches depending on the problem instance [19]. This section therefore also provides an analysis to reveal the statistical significance of the obtained results. An open question for researchers is whether there is a hegemonic strategy or heuristic class that could be used for the majority of problem instances.

As part of a thorough and detailed assessment, the 80 different MSA strategies within the framework were evaluated using two of the most frequently referenced protein sequence benchmarks: BAliBASE [27] and HOMSTRAD [28]. The accuracies of the alignments generated for both benchmarks were measured using a score, calculated with BAliBASE's own tool, which is referred to here as the Developer Score (DS) [4] to avoid any confusion with the earlier SP score definition. This tool counts the number of times the position of each pair of aligned residues in an MSA matches the corresponding residue pair and position in the correctly aligned reference MSA. To obtain a normalized value, the count is divided by the total number of aligned residue pairs in the reference alignment. Note, that the DS only considers residues pairs of the columns in the reference alignment where the number of gaps is less than 20%.

As noted previously in [19, 20, 23], the qualities of alignments, as measured by the DS, tend to be higher for sequences with a high degree of similarity. With this in mind, the benchmark problem instances have been subdivided into three groups according to their Identity values.

The remainder of this section first describes some of the characteristics of the two chosen benchmarks in Subsections 4.1 and 4.2, respectively. This is followed by two statistical evaluations, Subsection 4.3 presents the benefits of Consistency Transformation and Iterative Refinement for Progressive Alignments while Subsection 4.4 compares the different techniques within each stage of Progressive Alignment heuristics.

Finally, while the aforementioned evaluation results might indicate which class of heuristics is likely to obtain superior alignments, Subsection 4.5 takes a different perspective when analyzing the performance of the different design options. It attempts to identify whether a given class is always dominated by another and how frequently each class finds the highest scoring alignment between them.

## 4.1 The BAliBASE benchmark

One of our motives is to evaluate the performance of MSA heuristics in relation to sequences with different degrees of Identity. While Reference Set 9 from the BAliBASE benchmark [27] consists of 4 subsets of alignments, only the first one, which is comprised of sequences with true positive linear motifs (important regions of proteins), comes separated into groups according to sequence Identity. A total of 84 separate MSA instances are divided into 3 groups:

1. **RV911** contains 29 distinct multiple alignment instances or BOXes. Each BOX is composed, on average, of 15 sequences, with less than <20% Identity between pairs. For this group, the average length of the sequences in a BOX ranges from 227 to 1474 residues.

2. **RV912** contains 28 distinct multiple alignment instances, each composed of, on average, 8 sequences, with identities of between 20% and 40% between pairs. For this group, the average sequence length in a BOX ranges from 98 to 978 residues.

3. **RV913** contains 27 distinct multiple alignment instances, each composed of, on average, 10 sequences, with identities of between 40% and 80% between pairs. For this group, the average length of the sequences in a BOX ranges from 99 to 1374 residues.

## 4.2 The HOMSTRAD benchmark

HOMSTRAD [28] is a curated database of structure-based alignments for homologous protein families. Although HOMSTRAD is not a benchmark per se, as it contains numerous alignments in its database, it is used as a benchmark by several authors to test the accuracy of their tools. HOMSTRAD is composed of sequence alignments using structure information extracted from the Protein Data Bank (PDB) [46] and sequence alignments from other databases such as Pfam [47] and SCOP [48]. While HOMSTRAD contains 1, 031 instances, most are comprised of only two sequences. In this paper, all of the 400 MSA instances with more than 2 sequences were used.

Differently from the selected BAliBASE set that has already been divided according to sequence Identity, as HOMSTRAD does not employ such a classification, we have grouped the instances according to the sequence Identity of their reference MSA based on Eq 2.

$$Id(A) = \left( \sum_{i=1}^{n-1} \sum_{j=i+1}^{n} \frac{\sum_{k=1}^{L} M(a_{ik}, a_{jk})}{L - G(i,j)} \right) \times \frac{100}{\binom{n}{2}} \qquad (2)$$

where $A$ is the MSA, $L$ is the length of the aligned sequences, $n$ is the number of sequences, $G(i, j)$ is the number of gap-only columns in the alignment of sequences $i$ and $j$. $M(a_{ik}, a_{jk})$ is defined to be 1 if residue $k$ of sequence $i$ ($a_{ik}$) is equal to residue $k$ of sequence $j$ ($a_{jk}$) and neither are gaps, otherwise, $M(a_{ik}, a_{jk})$ is 0.

Based on the Identity values, the corresponding HOMSTRAD problem instances were also divided into 3 groups, using the same criteria adopted by the BAliBASE, and have the following characteristics:

1. **G1** contains 73 distinct MSA instances. Each instance is composed, on average, of 6 sequences, and has a sequence Identity percentage of less than <20%. For this group, the average length of the sequences in each problem instance ranged from 36 to 723 residues.

2. **G2** contains 209 distinct MSA instances, each composed of, on average, 6 sequences, and with a sequence Identity percentage between 20% and 40%. For this group, the average sequence length in each instance ranged from 14 to 859 residues.

3. **G3** contains 118 distinct MSA instances, each composed of, on average, 5 sequences, with a sequence Identity percentage between 40% and 80%. For this group, the average sequence length in each instance ranged from 23 to 861 residues.

## 4.3 Impact of consistency transformation and iterative refinement on progressive alignments

The principal aim of this paper is to evaluate the impacts that Consistency Transformation and Iterative Refinement have on the qualities of alignments produced by Progressive Alignment schemes. The relative qualities of the alignments produced by the 80 different MSA strategies (Subsection 3.6) were compared based on their respective Developer Scores for each of the 484 problem instances taken from the BAliBASE and HOMSTRAD benchmarks. Given that one MSA tool may produce a higher quality alignment than another tool for one instance and a lower one for another, it is common to compare tools based on their average accuracy (DS values) [22, 23, 34, 37] or determine the statistical significance of the differences between scores obtained by the strategies being compared [30].

To determine the latter, it was first necessary to verify whether the Developer Scores obtained by each of the 80 MSA strategies for a given benchmark group of problem instances followed a normal distribution in order to identify whether a parametric or non-parametric test should be used. Given the two benchmarks were each divided into 3 groups, and the 80 strategies, a total of 480 samples (2 benchmarks × 3 groups × 80 strategies) were evaluated. The frequencies of the p-values obtained by a Shapiro test for each sample are shown in Fig 1. The results show that 259 of the 480 samples (≈53.9%) have p-values lower than 0.05 and thus, with 95% probability, these don't follow a normal distribution. Thus, given the high probability that at least one sample does not follow a normal distribution, the non-parametric Wilcoxon signed-rank test [29] was chosen to compare the different heuristic classes.

In the context of this Wilcoxon test, if one wishes to determine if a given variant A obtains alignments with better Developer Scores than another variant B, it is necessary to refute the *null hypothesis* that assumes the scores of variant B are greater or equal to the corresponding ones of variant A, i.e., the opposite of our *alternative hypothesis*. The *p-value* derived from the test is the probability that the *null hypothesis* is true. Should this p-value be higher than a predetermined threshold (by convention, $\geq 0.05$), one can conclude that the samples do not provide sufficient evidence that variant A has higher scores than B. Otherwise one can refute the *null hypothesis*, as the chance of variant A obtaining a higher score than variant B is considered to be *statistically significant* [29].

The experimental analysis focuses on comparing the following four heuristic classes, with 20 implementation variations for each: Progressive Alignment (**P**), composed of the 3 stages of this classic strategy; Consistency-based Alignment (**C**) that uses additional information to

**Shapiro p-values frequency**

**Fig 1. The distribution of the Shapiro test p-values for all 480 samples.**

improve the final stage of the progressive strategy; Iterative Alignment (**I**) that, in an additional stage, makes a number of attempts (in these experiments, limited to a maximum of 100 iterations) to refine the alignment produced by the corresponding progressive strategy, and; the Consistency-based Progressive Alignment with Iterative Refinement (**CI**), where the iterative stage follows the generation of a Consistency-based progressive alignment.

Our evaluation is first summarized over three pairs of tables, Tables 2 to 7, with each table containing p-values for Wilcoxon signed-rank tests related to one of the six problem instance groups taken from the two benchmarks, HOMSTRAD and BAliBASE. Initially, Tables 2 and 3 present comparative results for the respective benchmarks groups with low Identity (i.e., the groups containing the problem instances with Identity percentages lower than 20%), Tables 4 and 5 show results for the groups where the instances have Identity percentages between 20% and 40%, and the last two (Tables 6 and 7) for the groups with instances that have Identity percentages greater than 40%.

Each table is composed of 7 columns, the first being a list of 20 variations of techniques used in the first (Subsection 3.2) and second stages (Subsection 3.1) of a Progressive Alignment. Each combination has been incorporated into a representative MSA strategy for each of the four heuristics classes (**P**, **C**, **I**, and **CI**). The corresponding variants of each heuristic class are then compared in a pairwise fashion using a Wilcoxon signed-rank test, based on the Developer Scores obtained for the set of problem instances in the benchmark group of that table. Thus, each table presents a further six columns with headings that identify the *alternative hypothesis* of the pairwise test. P-values less than 0.05 (highlighted in bold) mean the *null*

 On closing the inopportune gap with consistency transformation and iterative refinement

**Table 2. Wilcoxon test applied to the Developer Scores for G1.**

|  | C > P | I > P | C > I | CI > P | CI > I | CI > C |
|---|---|---|---|---|---|---|
| F.NJ | **0.00114** | 0.29929 | **0.00599** | **0.00185** | **0.00691** | 0.42319 |
| F.SLMAX | **0.00304** | 0.28054 | **0.01291** | **0.00207** | **0.01239** | 0.50390 |
| F.SLMIN | **0.00315** | 0.35647 | **0.00835** | **0.00257** | **0.00943** | 0.50234 |
| F.UPGMA | **0.00304** | 0.28584 | **0.01736** | **0.00254** | **0.01822** | 0.48907 |
| K.NJ | **8.30e-5** | 0.08205 | **0.00619** | **2.70e-5** | **0.00171** | 0.68628 |
| K.SLMAX | **5.10e-5** | 0.06642 | **0.00442** | **2.90e-5** | **0.00164** | 0.67298 |
| K.SLMIN | **8.10e-5** | 0.09690 | **0.00378** | **4.30e-5** | **0.00149** | 0.63032 |
| K.UPGMA | **6.90e-5** | 0.07600 | **0.00530** | **4.40e-5** | **0.00190** | 0.64790 |
| L.NJ | **2.60e-5** | 0.08234 | **0.00236** | **2.70e-5** | **0.00138** | 0.57987 |
| L.SLMAX | **1.80e-5** | 0.15443 | **0.00082** | **2.30e-5** | **0.00069** | 0.52730 |
| L.SLMIN | **0.00012** | 0.22677 | **0.00119** | **4.60e-5** | **0.00047** | 0.60945 |
| L.UPGMA | **0.00011** | 0.25856 | **0.00096** | **7.20e-5** | **0.00055** | 0.57067 |
| P.NJ | **9.60e-8** | **0.00762** | **0.00169** | **1.70e-8** | **0.00025** | 0.69456 |
| P.SLMAX | **6.90e-8** | **0.01686** | **0.00023** | **1.20e-8** | **3.20e-5** | 0.73507 |
| P.SLMIN | **1.00e-7** | 0.08659 | **3.10e-5** | **3.70e-8** | **7.90e-6** | 0.63548 |
| P.UPGMA | **7.00e-8** | **0.03483** | **5.40e-5** | **2.00e-8** | **1.50e-5** | 0.67369 |
| Q.NJ | **0.00053** | 0.15073 | **0.01507** | **0.00085** | **0.01606** | 0.47114 |
| Q.SLMAX | **0.00373** | 0.26751 | **0.01839** | **0.00342** | **0.01559** | 0.49688 |
| Q.SLMIN | **0.00592** | 0.28518 | **0.01686** | **0.00434** | **0.01208** | 0.51327 |
| Q.UPGMA | **0.00599** | 0.26494 | **0.02111** | **0.00579** | **0.02131** | 0.49141 |

**Table 3. Wilcoxon test applied to the Developer Scores for RV911.**

|  | C > P | I > P | C > I | CI > P | CI > I | CI > C |
|---|---|---|---|---|---|---|
| F.NJ | **0.00293** | 0.34298 | **0.00628** | **0.00383** | **0.00670** | 0.50621 |
| F.SLMAX | **0.00700** | 0.44742 | **0.00943** | **0.00692** | **0.00983** | 0.49069 |
| F.SLMIN | **0.00867** | 0.45974 | **0.00924** | **0.00757** | **0.00796** | 0.47519 |
| F.UPGMA | **0.02160** | 0.53409 | **0.02160** | **0.02042** | **0.02042** | 0.48449 |
| K.NJ | **8.12e-5** | 0.15050 | **0.00096** | **0.00011** | **0.00128** | 0.48139 |
| K.SLMAX | **5.38e-6** | 0.26951 | **4.03e-5** | **8.84e-6** | **5.92e-5** | 0.47519 |
| K.SLMIN | **7.87e-5** | 0.34870 | **0.00029** | **6.52e-5** | **0.00021** | 0.45357 |
| K.UPGMA | **2.80e-6** | 0.29309 | **1.95e-5** | **4.18e-6** | **2.91e-5** | 0.46591 |
| L.NJ | **0.00128** | 0.25930 | **0.00429** | **0.00098** | **0.00302** | 0.44433 |
| L.SLMAX | **0.00021** | 0.28246 | **0.00089** | **0.00013** | **0.00055** | 0.41988 |
| L.SLMIN | **0.00115** | 0.36611 | **0.00262** | **0.00086** | **0.00269** | 0.46591 |
| L.UPGMA | **0.00227** | 0.37786 | **0.00460** | **0.00177** | **0.00374** | 0.44741 |
| P.NJ | **5.86e-7** | 0.19190 | **7.67e-6** | **5.65e-7** | **5.99e-6** | 0.39571 |
| P.SLMAX | **1.62e-7** | 0.12329 | **3.36e-6** | **1.32e-7** | **2.16e-6** | 0.39573 |
| P.SLMIN | **8.67e-7** | 0.35159 | **5.18e-7** | **5.01e-7** | **1.05e-6** | 0.43820 |
| P.UPGMA | **6.10e-7** | 0.41685 | **1.48e-6** | **5.22e-7** | **1.18e-6** | 0.46901 |
| Q.NJ | **0.00091** | 0.30118 | **0.00310** | **0.00064** | **0.00295** | 0.47829 |
| Q.SLMAX | **0.00256** | 0.43208 | **0.00401** | **0.00211** | **0.00340** | 0.47519 |
| Q.SLMIN | **0.00239** | 0.50931 | **0.00282** | **0.00244** | **0.00250** | 0.49690 |
| Q.UPGMA | **0.00562** | 0.41079 | **0.00963** | **0.00439** | **0.00747** | 0.45973 |

**Table 4. Wilcoxon test applied to the Developer Scores for G2.**

|          | C > P    | I > P    | C > I   | CI > P   | CI > I  | CI > C  |
|----------|----------|----------|---------|----------|---------|---------|
| F.NJ     | **0.03341** | 0.21291 | 0.14962 | **0.03866** | 0.15189 | 0.50485 |
| F.SLMAX  | **0.04377** | 0.36248 | 0.09070 | **0.06118** | 0.10620 | 0.53983 |
| F.SLMIN  | **0.04251** | 0.33346 | 0.10088 | **0.05972** | 0.12503 | 0.54705 |
| F.UPGMA  | **0.04916** | 0.36097 | 0.10038 | **0.06545** | 0.11696 | 0.53356 |
| K.NJ     | **0.00445** | 0.12737 | 0.06291 | **0.00498** | 0.07176 | 0.50953 |
| K.SLMAX  | **0.00553** | 0.16280 | 0.05368 | **0.00724** | 0.05963 | 0.52405 |
| K.SLMIN  | **0.00360** | 0.14551 | **0.04491** | **0.00573** | 0.06157 | 0.54914 |
| K.UPGMA  | **0.00331** | 0.12881 | **0.04769** | **0.00499** | 0.06152 | 0.54063 |
| L.NJ     | **0.01553** | 0.26941 | 0.06392 | **0.01810** | 0.06919 | 0.50194 |
| L.SLMAX  | **0.01652** | 0.23630 | 0.07830 | **0.02010** | 0.08345 | 0.51114 |
| L.SLMIN  | **0.01879** | 0.29792 | 0.06201 | **0.02440** | 0.07083 | 0.53001 |
| L.UPGMA  | **0.01697** | 0.27994 | 0.06098 | **0.02090** | 0.06979 | 0.52131 |
| P.NJ     | **1.10e-5** | **0.00101** | 0.12519 | **3.73e-6** | 0.07648 | 0.37777 |
| P.SLMAX  | **2.00e-6** | **0.00179** | **0.04067** | **5.52e-7** | **0.02103** | 0.38440 |
| P.SLMIN  | **5.60e-6** | **0.02673** | **0.00700** | **3.71e-6** | **0.00530** | 0.44478 |
| P.UPGMA  | **1.50e-6** | **0.00556** | **0.01665** | **5.17e-7** | **0.00976** | 0.41473 |
| Q.NJ     | **0.02736** | 0.17556 | 0.17400 | **0.03234** | 0.17851 | 0.50711 |
| Q.SLMAX  | **0.03335** | 0.28008 | 0.10196 | **0.04721** | 0.12411 | 0.55634 |
| Q.SLMIN  | **0.03593** | 0.32395 | 0.08886 | 0.05285 | 0.11072 | 0.56815 |
| Q.UPGMA  | **0.03427** | 0.30314 | 0.09603 | **0.04638** | 0.11057 | 0.54401 |

**Table 5. Wilcoxon test applied to the Developer Scores for RV912.**

|          | C > P    | I > P    | C > I   | CI > P   | CI > I  | CI > C  |
|----------|----------|----------|---------|----------|---------|---------|
| F.NJ     | **0.03837** | 0.39025 | 0.05879 | **0.03444** | 0.05148 | 0.47386 |
| F.SLMAX  | **0.02759** | 0.46408 | **0.03202** | **0.03086** | **0.03508** | 0.51308 |
| F.SLMIN  | **0.02148** | 0.43812 | **0.02759** | **0.02812** | **0.03445** | 0.54243 |
| F.UPGMA  | **0.02974** | 0.47060 | **0.03321** | **0.03382** | **0.03572** | 0.53267 |
| K.NJ     | **0.00894** | 0.54568 | **0.00782** | **0.00875** | **0.00782** | 0.49673 |
| K.SLMAX  | **0.01691** | 0.49019 | **0.01691** | **0.01656** | **0.01725** | 0.50000 |
| K.SLMIN  | **0.02415** | 0.48366 | **0.02811** | **0.02509** | **0.02973** | 0.51961 |
| K.UPGMA  | **0.01434** | 0.52287 | **0.01319** | **0.01464** | **0.01464** | 0.51308 |
| L.NJ     | **0.01590** | 0.49346 | **0.01796** | **0.01870** | **0.02107** | 0.52289 |
| L.SLMAX  | **0.02149** | 0.50981 | **0.02279** | **0.02147** | **0.02368** | 0.48692 |
| L.SLMIN  | **0.03906** | 0.48039 | **0.04978** | **0.03382** | **0.04339** | 0.49673 |
| L.UPGMA  | **0.02323** | 0.50000 | **0.02558** | **0.02191** | **0.02557** | 0.50981 |
| P.NJ     | **0.00235** | 0.39973 | **0.00247** | **0.00212** | **0.00218** | 0.47060 |
| P.SLMAX  | **0.00030** | 0.36844 | **0.00048** | **0.00019** | **0.00038** | 0.46408 |
| P.SLMIN  | **0.00098** | 0.32608 | **0.00122** | **0.00066** | **0.00092** | 0.49346 |
| P.UPGMA  | **0.00030** | 0.33797 | **0.00044** | **0.00024** | **0.00041** | 0.48692 |
| Q.NJ     | **0.02279** | 0.50000 | **0.02368** | **0.02149** | **0.02192** | 0.46734 |
| Q.SLMAX  | **0.01066** | 0.43813 | **0.01465** | **0.01265** | **0.01833** | 0.52940 |
| Q.SLMIN  | **0.03262** | 0.53266 | **0.03573** | **0.03384** | **0.03572** | 0.52940 |
| Q.UPGMA  | **0.01908** | 0.51634 | **0.01833** | **0.02025** | **0.02025** | 0.51962 |

**Table 6. Wilcoxon test applied to the Developer Scores for G3.**

|  | C > P | I > P | C > I | CI > P | CI > I | CI > C |
|---|---|---|---|---|---|---|
| F.NJ | 0.30899 | 0.50114 | 0.30430 | 0.32705 | 0.32327 | 0.52286 |
| F.SLMAX | 0.32327 | 0.41677 | 0.39748 | 0.35900 | 0.42910 | 0.53237 |
| F.SLMIN | 0.32705 | 0.40822 | 0.41490 | 0.34799 | 0.42685 | 0.51258 |
| F.UPGMA | 0.31745 | 0.41193 | 0.38903 | 0.34940 | 0.41788 | 0.52857 |
| K.NJ | 0.28060 | 0.45019 | 0.31102 | 0.29296 | 0.31882 | 0.50534 |
| K.SLMAX | 0.27292 | 0.44755 | 0.29500 | 0.29001 | 0.31034 | 0.51258 |
| K.SLMIN | 0.28545 | 0.45208 | 0.31610 | 0.29428 | 0.32190 | 0.49962 |
| K.UPGMA | 0.27547 | 0.45625 | 0.29698 | 0.27547 | 0.29500 | 0.49009 |
| L.NJ | 0.32153 | 0.51144 | 0.30160 | 0.38244 | 0.35720 | 0.57203 |
| L.SLMAX | 0.29728 | 0.45095 | 0.33430 | 0.36903 | 0.40819 | 0.59069 |
| L.SLMIN | 0.32736 | 0.45738 | 0.36471 | 0.37698 | 0.41675 | 0.56866 |
| L.UPGMA | 0.31267 | 0.45057 | 0.35044 | 0.37880 | 0.41526 | 0.58362 |
| P.NJ | 0.05057 | 0.25374 | 0.16091 | 0.05522 | 0.16212 | 0.51372 |
| P.SLMAX | **0.03383** | 0.25623 | 0.09584 | **0.03166** | 0.09013 | 0.49428 |
| P.SLMIN | **0.03153** | 0.23696 | 0.10474 | **0.03000** | 0.09930 | 0.50534 |
| P.UPGMA | **0.03276** | 0.24586 | 0.09663 | **0.02822** | 0.08302 | 0.48742 |
| Q.NJ | 0.28098 | 0.48133 | 0.28747 | 0.30731 | 0.31814 | 0.53237 |
| Q.SLMAX | 0.30362 | 0.45929 | 0.33850 | 0.34940 | 0.38465 | 0.55283 |
| Q.SLMIN | 0.31677 | 0.45285 | 0.35472 | 0.35118 | 0.39160 | 0.53996 |
| Q.UPGMA | 0.28195 | 0.44265 | 0.32326 | 0.32843 | 0.37483 | 0.56038 |

**Table 7. Wilcoxon test applied to the Developer Scores for RV913.**

|  | C > P | I > P | C > I | CI > P | CI > I | CI > C |
|---|---|---|---|---|---|---|
| F.NJ | 0.34214 | 0.53793 | 0.32018 | 0.33581 | 0.31095 | 0.49655 |
| F.SLMAX | 0.35495 | 0.60243 | 0.28693 | 0.36466 | 0.29884 | 0.51726 |
| F.SLMIN | 0.37771 | 0.57550 | 0.30182 | 0.37443 | 0.29881 | 0.50000 |
| F.UPGMA | 0.39760 | 0.57551 | 0.33265 | 0.41437 | 0.34531 | 0.51382 |
| K.NJ | 0.30488 | 0.56191 | 0.28400 | 0.29286 | 0.26951 | 0.48619 |
| K.SLMAX | 0.18644 | 0.51381 | 0.17956 | 0.18877 | 0.18413 | 0.50345 |
| K.SLMIN | 0.23903 | 0.51381 | 0.24173 | 0.26100 | 0.26384 | 0.50691 |
| K.UPGMA | 0.25265 | 0.54480 | 0.24173 | 0.28105 | 0.27239 | 0.53105 |
| L.NJ | 0.41438 | 0.51381 | 0.39094 | 0.38432 | 0.36139 | 0.46551 |
| L.SLMAX | 0.35173 | 0.58229 | 0.25263 | 0.35817 | 0.25818 | 0.51726 |
| L.SLMIN | 0.38763 | 0.57551 | 0.32639 | 0.38432 | 0.32328 | 0.50346 |
| L.UPGMA | 0.34852 | 0.55848 | 0.28692 | 0.34852 | 0.28692 | 0.50345 |
| P.NJ | **0.02190** | 0.47929 | **0.02283** | **0.02283** | **0.02378** | 0.51727 |
| P.SLMAX | **0.01369** | 0.47240 | **0.01430** | **0.01369** | **0.01494** | 0.50345 |
| P.SLMIN | **0.03332** | 0.48964 | **0.03463** | **0.03397** | **0.03598** | 0.50691 |
| P.UPGMA | **0.01595** | 0.50000 | **0.01737** | **0.01595** | **0.01737** | 0.50345 |
| Q.NJ | 0.34851 | 0.45862 | 0.34851 | 0.34214 | 0.34214 | 0.48619 |
| Q.SLMAX | 0.40430 | 0.61239 | 0.29286 | 0.40766 | 0.29584 | 0.50345 |
| Q.SLMIN | 0.41101 | 0.54823 | 0.33898 | 0.40765 | 0.33582 | 0.49655 |
| Q.UPGMA | 0.38102 | 0.60907 | 0.26952 | 0.38432 | 0.27526 | 0.50346 |

*hypothesis* can be rejected, affirming the condition of the column heading: **C** > **P** indicates which Consistency-based variants are statistically better than their Progressive-only ones; **I** > **P**, which Iterative variants are better than their Progressive-only ones; **C** > **I**, which Consistency-based variants are better than Iterative ones; and similarly for the columns **CI** > **P**, **CI** > **I** and **CI** > **C**, which Consistency-based Progressive Alignment with Iterative Refinement variants are better than their respective Progressive, Iterative and Consistency-based approaches.

When considering alignments for the problem instances with low Identity (groups **RV911** and **G1**), the results presented in Tables 2 and 3 highlight pretty clearly that MSA heuristics that employ Consistency Transformation (**C** and **CI**), in general, are better than both Progressive Alignment (**C** > **P** and **CI** > **P**) and Iterative Alignment (**C** > **I** and **CI** > **I**), given that the p-values in the four columns, in both tables, for the corresponding variants are all less than 0.05. Unlike the relation to the other heuristics, the test results for the heuristics combining consistency with Iterative Refinement (**CI**) were similar to those obtained by the Consistency-based heuristics (**C**), so much so that the differences between the respective pairs were not statistically significant, as seen in the columns **CI** > **C**. The qualities of the alignments varied depending on the variant strategy and problem instance, without revealing any specific trend as to which heuristic class is better.

Furthermore, Iterative Refinement (**I**), an alternative approach to address the issue of inopportune gaps in progressive alignments, also does not appear to have variants with performances that are statistically significant in relation to their **P** variants, i.e., the distribution of scores refutes the hypothesis that **I** > **P**. In fact, only 3 of the four PROBA variants of the iterative strategy (PROBA with the Guide tree construction algorithms NJ, SLMAX, and UPGMA, respectively) can claim to have improvements over their corresponding Progressive Alignment counterparts that are statistically relevant, and which only occurred for the group **G1**.

Tables 4 and 5 present the Wilcoxon test results for the groups, **RV912** and **G2**, with Identity percentages between 20% and 40%. The p-values already indicate a change in tendencies with respect to the previous groups, in particular **G2** (note the fewer bold numbers compared with Tables 2 and 3). While all 20 Consistency-based variants continue to be statistically better than their progressive alignment ones for both instance groups, the quality of the alignments they produce when compared to those of the Iterative refinement heuristics (see column **C** > **I**) were statistically significant for only 5 of 20 variants, for the HOMSTRAD group, and 19 of 20 for BAliBASE. The stark difference between these two groups may in part be due to the distribution of Identity values within each group. While 92.86% of instances in **RV912** have Identity values below 30%, in **G2** only 47.85% have Identities below the same threshold.

The heuristics that combine Consistency Transformation and Iterative Refinement (**CI**) exhibit a somewhat similar behavior to the Consistency-based ones, when compared to both their respective Progressive Alignment (**CI** > **P**) and Iterative Refinement (**CI** > **I**) equivalents. However, the number of variants for which the differences in alignment quality were statistically significant drop slightly to 3 for **G2** and stays the same, 19, for **RV912**. Comparing **CI** with **C** gives rise to p-values around 0.5, therefore, again, statistically there appears to be little benefit in combining these two techniques. Finally, only in the case of **G2**, can it be said that the technique of Iterative Refinement improved the qualities of Progressive Alignments that employ the scoring technique PROBA.

The last two groups of Wilcoxon test results in Tables 6 and 7) are for instances with Identity percentages greater than 40%. For these types of sequences, no particular heuristic class is better than the others, with statistical significance, for any variation. As indicated by the relatively few p-values in bold, the tests show that statistically significant improvements were only

seen for Consistency-based variants that use PROBA (except with NJ) over their corresponding progressive aligners in the case of group **G3**, and over both their corresponding progressive and iterative aligners in the case of group **RV913**.

In summary, based on pairwise Wilcoxon signed-rank tests to identify a dominant heuristic class, we find (with 95% confidence) that, for Identity values between 40% and 80%, in general, there is statistically no significant difference between corresponding implementations of the four heuristic classes: Progressive Alignment (**P**); Consistency-based Alignment (**C**); Iterative Alignment (**I**), and; the Consistency-based Progressive Alignment with Iterative Refinement (**CI**). As the Identity values decrease, Consistency-based progressive alignments (**C** and **CI**) begin to achieve statistically significant improvements over both Progressive **P** and Iterative **I** Alignments. However, independent of Identity values, the addition of an Iterative stage to the MSA pipeline of both **P** and **C** classes of heuristics does not appear to be statistically beneficial. As mentioned earlier, one reason for this is due to the difference between the function WSP that is being optimized by the Iterative Refinement technique and the DS used to measure alignment quality. We have observed several instances where improvements in the WSP score during the iterative stage even led to worse Developer Scores.

## 4.4 The impact of similarity scoring and guide tree generation techniques on Progressive-based Alignments

Following a similar line of investigation as the previous section, one might be interested in knowing which are the best techniques for calculating similarity scores or for generating guide trees, in general. A preliminary analysis of these techniques was presented in [19] and to complement that work with a statistical analysis, this section uses the same methodology presented in Section 4.3.

To compare the 5 similarity scoring techniques, the Developer Scores (DS) obtained by each technique, for each of the 6 benchmark Identity groups, were gathered to form 6 sets of samples for that technique irrespective of the combination of guide tree technique and remaining stages used by the strategy.

Table 8 presents the Wilcoxon test results comparing the 5 techniques (FULL, KMERS, LCS, QUICK and PROBA), represented by their initial letter. As shown previously in Subsection 4.3, values in bold indicate the cases where the null hypothesis was rejected. In these cases,

**Table 8. Wilcoxon test to compare similarity scoring techniques.**

|         | F > K       | F > L       | F > P        | F > Q    | K > P       |
|---------|-------------|-------------|--------------|----------|-------------|
| G1      | **0.00239** | **0.01106** | **4.70e-11** | 0.44092  | **1.28e-5** |
| G2      | **0.03151** | 0.21076     | **1.31e-11** | 0.39444  | **7.68e-7** |
| G3      | 0.45689     | 0.59746     | **0.00246**  | 0.49723  | **0.00373** |
| RV911   | **1.73e-6** | **0.0043**  | **6.99e-13** | 0.10243  | **0.00142** |
| RV912   | 0.12054     | 0.26368     | **0.00013**  | 0.42194  | **0.00533** |
| RV913   | 0.24945     | 0.44931     | **0.00004**  | 0.45232  | **0.00023** |
|         | L > K       | L > P       | Q > K        | Q > L    | Q > P       |
| G1      | 0.33175     | **3.46e-6** | **0.00511**  | **0.01819** | **2.13e-10** |
| G2      | 0.11721     | **3.04e-10**| 0.05470      | 0.31272  | **8.88e-11** |
| G3      | 0.37968     | **0.00117** | 0.47667      | 0.59894  | **0.00287** |
| RV911   | **0.04065** | **2.38e-6** | **0.00037**  | 0.07181  | **7.68e-10** |
| RV912   | 0.26469     | **0.00069** | 0.17995      | 0.33926  | **0.00051** |
| RV913   | 0.26848     | **0.00007** | 0.27607      | 0.50261  | **0.00008** |

**Table 9. Wilcoxon test to compare guide tree generation techniques.**

|  | NJ > MAX | NJ > MIN | NJ > UPG | MAX > MIN | MAX > UPG | MIN > UPG |
|---|---|---|---|---|---|---|
| G1 | 0.62064 | 0.74468 | 0.73032 | 0.63280 | 0.62597 | 0.49377 |
| G2 | 0.41524 | 0.55356 | 0.42957 | 0.63601 | 0.51379 | 0.37595 |
| G3 | 0.44523 | 0.46304 | 0.43036 | 0.51789 | 0.48532 | 0.46719 |
| RV911 | 0.37448 | 0.71471 | 0.75173 | 0.80100 | 0.82343 | 0.54210 |
| RV912 | 0.47616 | 0.68403 | 0.57262 | 0.70955 | 0.60335 | 0.37129 |
| RV913 | 0.48073 | 0.55148 | 0.54623 | 0.58000 | 0.56133 | 0.49627 |

the statement described in the column header is true for the given benchmark group. For example, the first p-value of 0.00239, being less than 0.05, indicates that FULL is better than KMERS (F > K) for HOMSTRAD G1.

The first observation is that PROBA is worse, with statistical significance, than the other four techniques, regardless of the benchmark group. For alignments with Identities of less than 20%, FULL is statistically better than three of the other techniques, the exception being QUICK, for both HOMSTRAD and BAliBASE. The same can be said about QUICK being better than KMERS and LCS for HOMSTRAD G1, and better than KMERS for BAliBASE RV911. Also, for RV911, LCS was better than KMERS and PROBA. For the other groups of the two benchmarks, the only other conclusion that can be drawn is that FULL is also better than KMERS for HOMSTRAD G2.

Analogous to Tables 8 and 9 presents the Wilcoxon test results for the pairwise comparison of the 4 guide tree generation techniques, Neighbor-Joining, UPGMA, SLMIN and SLMAX, represented in the table by NJ, UPG, MIN and MAX, respectively. As all the p-values in the table are above 0.05, none of the guide tree generation techniques are statistically superior to any of the others, indicating that the choosing the best guide tree generation technique for a given MSA problem instance might be a difficult task.

In summary, with the exception of PROBA, which is statistically a worse performer than the other 4 similarity scoring techniques, only in specific scenarios did a given technique stand out over the others. FULL and QUICK, both from the original Clustal W implementation, were only better for alignments with low identities. No such conclusions could be drawn for the guide tree generation techniques. These results corroborate the difficulty of choosing a single strategy to try to obtain the best alignments and that the use of a tool capable of testing different techniques, variations and strategies might be beneficial to the user.

## 4.5 Analyzing the top performances by heuristic class

As with any heuristic, there is no guarantee that it will always find the optimal solution. As the previous subsection suggests, although based on perhaps a relatively small set of instances, identifying a unique heuristic that consistently finds the best alignments is not easy. Even when Identity values are below 40%, although Consistency-based heuristics might obtain statistically better quality alignments for most problem instances, this does not mean that they find the best alignment for every instance. Rather than present average Developer Scores, this subsection analyses the frequency with which each class of heuristic obtained the best score that was found.

The percentage of times the best DS was obtained by each of the four heuristic classes (**P**, **C**, **I**, and **CI**), regardless of which variant was used, can be seen in Figs 2 to 7, in relation to the 6 respective benchmark groups. The blue-colored portion represents the percentage of times that a given class obtained this best score but was equaled by a variant of another

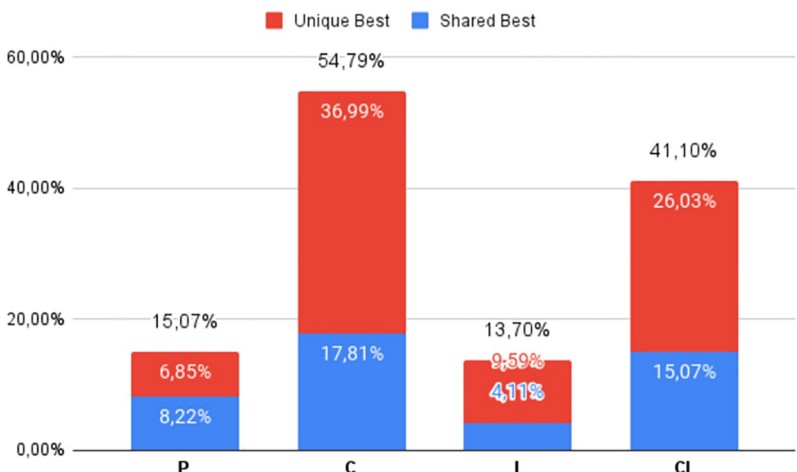

**Fig 2. G1: Percentage of unique best and tied best alignments per class.**

class. The red portion, however, represents the percentage of times that this class was the only one to obtain the highest score. The percentage values in black are the sum of these two values.

The frequencies of top performers for the groups of instances in the lowest Identity range are shown in Figs 2 and 3. First, notice that the distributions are different for the two benchmarks. As expected, **C** and **CI** obtain the best Developer Scores most of the time. But of note for **G1** (and true for **G2** and **G3** too) is the high frequency with which only one class is able to find the best alignment, almost 80%, with all four classes contributing to this statistic. Furthermore, the relatively novel and less explored class **CI** found improved alignments in a quite significant 26% of the instances in **G1** and 34% of **RV911** when compared with more traditional approaches. While **C** found the best alignment in 54.79% of the **G1** cases, it was bettered by **P** and **I** a combined, and not insignificant, 16.44% of the time. In the case of **RV911**, **CI** was the best overall performer finding the best score 75.86% of the time, while tying with **C** in 12 of 29

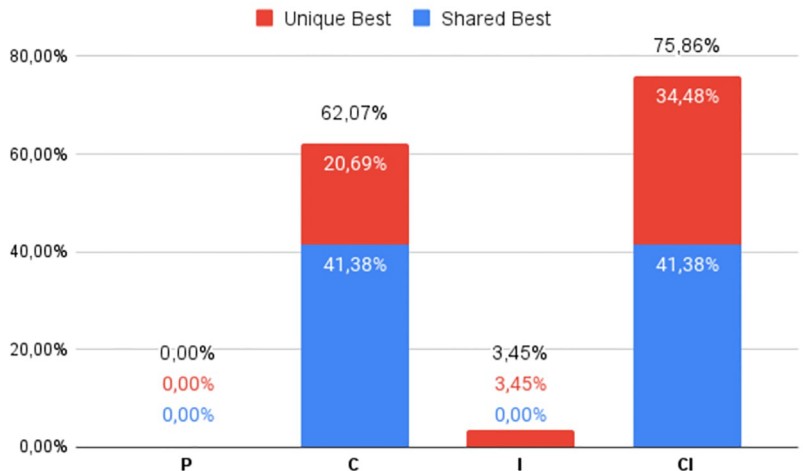

**Fig 3. RV911: Percentage of unique best and tied best alignments per class.**

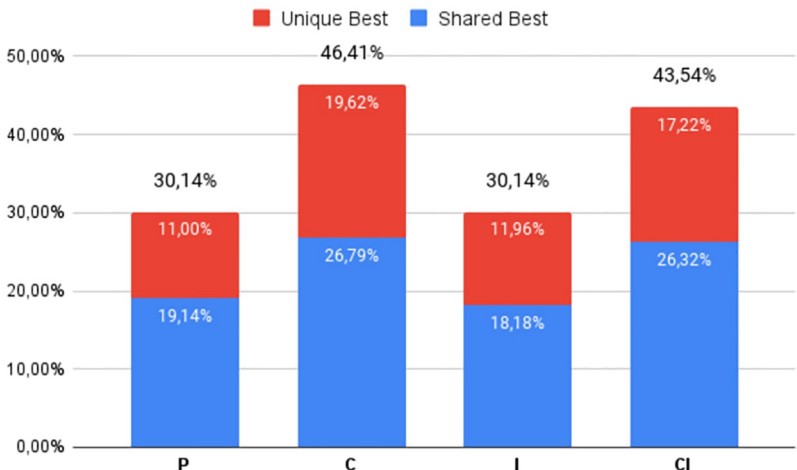

**Fig 4. G2: Percentage of unique best and tied best alignments per class.**

instances. Both **P** and **I** performed poorly in terms of this metric. While no Progressive Alignment obtained the best score, Iterative Alignment did uniquely find the best alignment for 1 of the instance problems.

An almost similar pattern occurs for the two groups with Identity percentages between 20% and 40%. For **G2** (Fig 4), there is a smaller disparity between all 4 classes, both in terms of overall top scores and unique top scores, although **C** and **CI** again have an advantage. Still, to be sure of finding the best alignments these results indicate the need to use more than one tool or a tool that implements all four classes. For **RV912** (Fig 5), the heuristics which use consistency

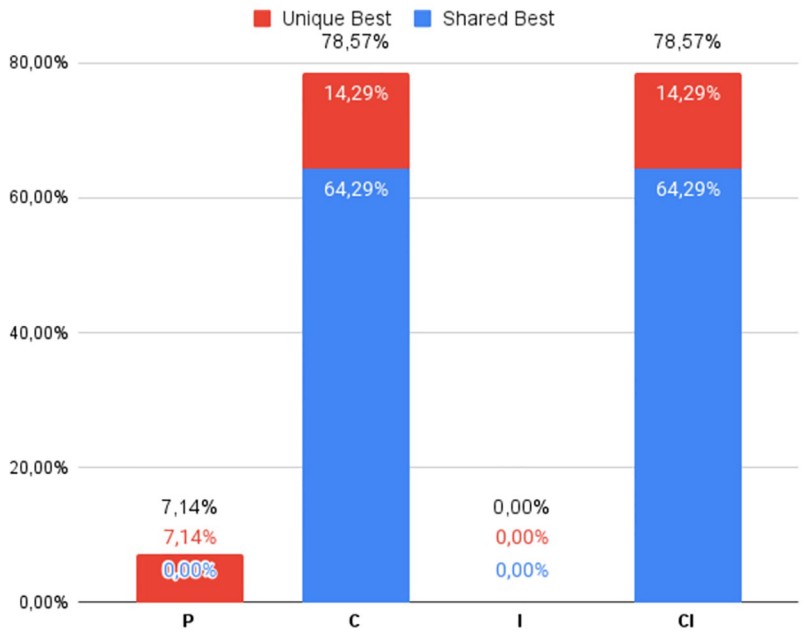

**Fig 5. RV912: Percentage of unique best and tied best alignments per class.**

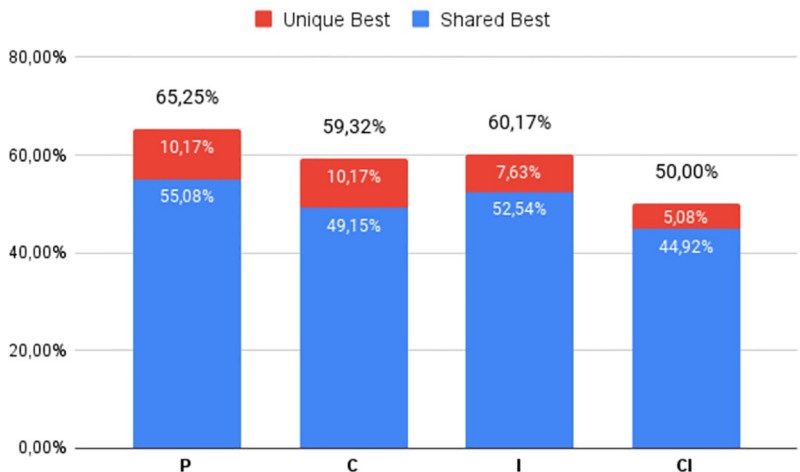

**Fig 6. G3: Percentage of unique best and tied best alignments per class.**

transformation (**C** and **CI**) again dominate, only losing out in two instances to the base progressive approach. Iterative Refinement has mixed success, again due to its scoring optimization: **CI** improved 4 instances and made 4 worse; **I** was not able to match or improve on the two top Progressive Alignments.

Finally, Figs 6 and 7 refer to the higher Identities value groups **G3** and **RV913**, where there is now a closer equilibrium between all four classes—a difference of just 15.25% for **G3** and 11.11% for **RV913**. This is further emphasized by the significant numbers of tied top scores. In the case of **G3**, the surprising fact is that the base progressive approach obtained the highest number of best scores. With respect to tied scores, when Identity values are above the twilight zone, both the **C** and **CI** approaches tend to be less effective, meaning simpler strategies can be equally successful.

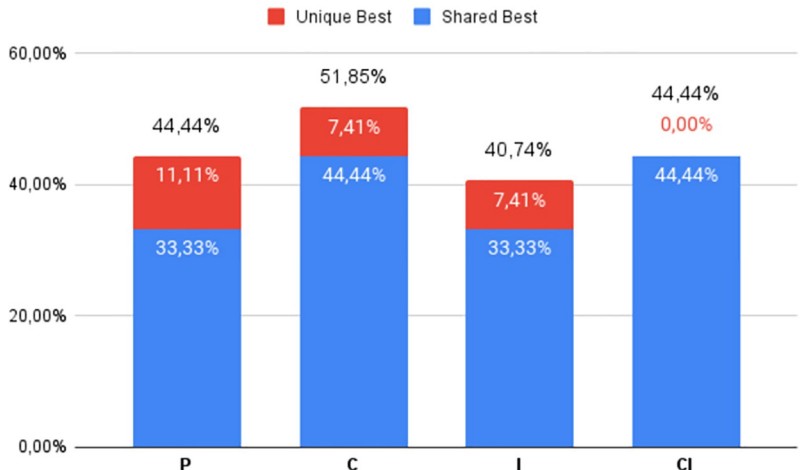

**Fig 7. RV913: Percentage of unique best and tied best alignments per class.**

## 5 Conclusions

Choosing the best tool to perform MSA is often difficult due to the diverse range of heuristics available and the variety of configuration options. Overcoming a disadvantage of the classical Progressive Alignment approach has seen the emergence of tools that use one of two techniques: Iterative Refinement or Consistency Transformation. Rather than propose a new tool or compare existing tools, this work focuses on understanding the impact of techniques, employed by successful tools, on alignment quality. Using an integrated framework [19], this paper presented a statistical evaluation of the alignment accuracy obtained by 80 combinations of existing techniques in order to evaluate the heuristic classes: Progressive Alignment; Progressive Alignment with Iterative Refinement; Consistency-based Alignment, and the lesser-known Consistency-based Alignment with Iterative Refinement. For sequences with a reasonable degree of similarity (above 40% Identity), all four heuristic classes are statistically comparable. Below this threshold, the two consistency-based heuristic classes stand out, while the benefits of adding an Iterative Refinement stage to either Progressive or Consistency-based Alignments are significantly smaller in comparison.

Given no single class of heuristic appears to be the silver bullet, our proposed framework aims to be able to explore a larger solution space by evaluating various, possibly novel, combinations of techniques simultaneously while re-utilizing common computations in order to find better solutions with less effort for a specific sequence alignment instance. While the focus of this work has not been to compare specific MSA tools, an indication of the **potential** of such a framework, in relation to its competitiveness with respect to 3 commonly adopted solutions: MUSCLE (that uses Iterative Refinement), ProbCons, and T-COFFEE (which employ Consistency Transformation) can be briefly summarized as follows. In relation to the HOMSTEAD benchmark groups, the three tools respectively only bettered the framework in 4.5%, 16%, and 16.75% of the 400 instances, and in 0%, 14.29%, and 8.33% of the 84 BAliBASE instances.

Ongoing work is continuing to integrate other techniques used by MSA tools and novel variants, in addition to a fifth class of MSA heuristic—regressive heuristics, while placing an increased focus on optimizing the MSA processing workflow and reducing execution times through the adoption of parallel and distributed computing. The use of information regarding both the structural and functional importance of each amino acid in proteins is seen as the gold standard to evaluate alignments. In the longer term, we plan to investigate how to extend and adapt our framework for the multiple structural alignment problem and to study its applicability in comparison with structural alignment tools such as TM-align [49].

## Author Contributions

**Conceptualization:** Mario João, Jr., Alexandre C. Sena, Vinod E. F. Rebello.

**Formal analysis:** Mario João, Jr.

**Investigation:** Mario João, Jr.

**Methodology:** Mario João, Jr., Alexandre C. Sena, Vinod E. F. Rebello.

**Software:** Mario João, Jr.

**Supervision:** Alexandre C. Sena, Vinod E. F. Rebello.

**Validation:** Mario João, Jr., Alexandre C. Sena, Vinod E. F. Rebello.

**Visualization:** Mario João, Jr.

**Writing – original draft:** Mario João, Jr., Alexandre C. Sena, Vinod E. F. Rebello.

**Writing – review & editing:** Alexandre C. Sena, Vinod E. F. Rebello.

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
