## [Decision Letter · Decision Letter 0]

10 Apr 2023

PONE-D-23-07871On closing the inopportune gap with consistency transformation and iterative refinementPLOS ONE

Dear Dr. João Junior,

Thank you for submitting your manuscript to PLOS ONE. After careful consideration, we feel that it has merit but does not fully meet PLOS ONE’s publication criteria as it currently stands. Therefore, we invite you to submit a revised version of the manuscript that addresses the points raised during the review process.

We look forward to receiving your revised manuscript.

Kind regards,

Yang Zhang

Academic Editor

PLOS ONE

Journal Requirements:

Reviewers' comments:

Reviewer's Responses to Questions

**Comments to the Author**

1. Is the manuscript technically sound, and do the data support the conclusions?

Reviewer #1: Partly

2. Has the statistical analysis been performed appropriately and rigorously? 

Reviewer #1: Yes

3. Have the authors made all data underlying the findings in their manuscript fully available?

Reviewer #1: No

4. Is the manuscript presented in an intelligible fashion and written in standard English?

Reviewer #1: Yes

5. Review Comments to the Author

Reviewer #1: Multiple sequence alignment (MSA) construction is actually a very important problem in bioinformatics and biology research. MSA has two slightly different meanings in recent studies. One is the traditional MSA definition that tries to align a set of sequences all against all to make the entire alignment reach an optimized global quality. The other is more like the definition of homologous sequences collection, which tries to detect and align the homologous sequences to a query sequence. In this paper, Mario Joao Jr. et. al., tried to focus on the first kind of MSA construction problem and assess the performance of different MSA construction methods, i.e., Progressive Alignments, Iterative Alignments, Consistency-based Alignments, and Consistency-based Progressive Alignments with Iterative Refinement, on two datasets and their six corresponding sub-datasets. Overall, the results shown in this paper are reasonable and logical, and the statistical analysis is appropriately made. However, there are still several issues that exist in the paper. Moreover, more comprehensive data analyses need to be done before the manuscript can be published.

Major:

1. In first paragraph of “Background section”, when describe the flow of Iterative Alignment, please describe how to split the whole alignments into two clearly, and how to combine the two split group profiles together as a new MSA/alignments.

2. In “Section 3.2”, please give a brief methodology description of UPGMA, NJ, SLMAX and SLMIN, especially the latter two, which are not commonly used in the field.

3. In “Section 4.3”, the authors mentioned that “The results of Shapiro tests for all of the 480 samples found that more than half of the samples did not follow a normal distribution, thus the non-parametric Wilcoxon signed-rank test was used to compare different heuristic classes.” Please show the data. For example, a histogram of the p-values from the Shapiro test. In addition, I think the number should be ‘484’, please recheck it.

4. In result analysis section, I think authors could show some comparison results between the 20 variations in the first column of each table, I think the readers are also curious about the performance of NJ vs UPGMA etc.

5. HOMSTRAD is a structure-based MSA database. Each sequence in a MSA has the PDB structure in HOMSTRAD database. Thus a structure-based analysis should be added to the paper, so that the readers could know how the performances of those 80 MSA construction methods are related to the structure alignment, since protein structures could directly reflect the protein evolution. For example, instead of just calculating the DS score for G1, G2 and G3, the authors could calculate the average TM-score of the pairwise proteins in a MSA based on the MSA alignment using TM-align/TM-score tool. Then show the comparisons of the average TM-scores for those 80 MSA construction methods.

Minor:

1. Multiple sequence alignments actually have two meanings in recent studies. One is its traditional definition, aligning the multiple sequences with each other, inserting reasonable gaps, and letting all sequences align well. This one is mainly used for phylogeny inference, evolution analysis, etc. The other one recently used in protein/RNA structure and function prediction is more like homologous sequence detection, which aligns all homologous to a 'query' sequence. This first one is the NP-hard problem, as the authors mentioned, while the second is mostly N times of pairwise alignments (N is the number of homologous sequences in MSA). In the “Introduction section”, please clearly mention the first one is what the authors focus on.

2. Recent co-evolutionary analyses are also based on MSA, like contact/distance prediction. In the “Introduction section”, please talk about this point.

3. In the first paragraph of “Section 4.3”, when you first mention “20 variations”, please cite “Section 3.1 and 3.2”. In the end of “Section 3”, please clearly say how you get 20 variations by combining 5 methods in “Section 3.1” and 4 methods in “Section 3.2”. Since when I first read the word “80 combinations” and “20 variations”, I am slightly confused where those methods came from.

4. Text in figure 4 is overlapping.

6. PLOS authors have the option to publish the peer review history of their article (what does this mean?). If published, this will include your full peer review and any attached files.

Reviewer #1: No

---

## [Author Response · Author response to Decision Letter 0]

30 May 2023

Major Points

1. In first paragraph of “Background section”, when describe the flow of Iterative Alignment, please describe how to split the whole alignments into two clearly, and how to combine the two split group profiles together as a

new MSA/alignments.

R.: To describe how the alignments are split, the following paragraph was added as the second paragraph of Section 2.

“While the method for dividing an alignment into two profiles in step (b) varies according to each tool, they all require a given criterion to be adopted to determine the allocation of each sequence to one of the respective profiles. The criteria most commonly used range from being simple (e.g., a random choice [30]), to something more elaborate (e.g., Tree Restricted Partitioning [31] presented in Section 3.5).”

2. In “Section 3.2”, please give a brief methodology description of UPGMA, NJ, SLMAX and SLMIN, especially the latter two, which are not commonly used in the field.

R.: A brief description of UPGMA, NJ, SLMAX, and SLMIN has been added at the end of Section 3.2. The following text added was:

“With the degree of similarity between pairs of sequences being held in a similarity matrix, the agglomerative UPGMA begins to construct its guide tree by representing each sequence as a single node and repeatedly joining the two nodes of the pair of sequences with the highest score in the similarity matrix to form a rooted tree. The similarity matrix is then altered to reflect the grouping of the two nodes into one. The two lines and columns of the similarity matrix, representing the two grouped nodes are substituted by a single line and a single column for the new root, with the scores between the remaining nodes being calculated as the average of the two scores being removed. The algorithm continues until all the nodes have

been combined into a single cluster, representing the guide tree. SLMIN and SLMAX follow the same algorithm, however, the new scores are no longer the average of the two scores but rather the minimum (in the case of SLMAX) or the maximum (for SLMIN) score between pairs. Similarly, Neighbor-Joining (NJ) is also a bottom-up agglomerative hierarchical clustering approach that uses a similarity matrix. Starting with a central node connected to all other nodes (sequences), the two least distant nodes (most similar sequences) are grouped, forming a tree that is connected to the central node. The distances between groups are calculated based on the path from the central node to each group, and the matrix is updated to incorporate the new group. This process is repeated until the completion of the guide tree.”

3. In “Section 4.3”, the authors mentioned that “The results of Shapiro tests for all of the 480 samples found that more than half of the samples did not follow a normal distribution, thus the non-parametric Wilcoxon signedrank test was used to compare different heuristic classes.” Please show the data. For example, a histogram of the p-values from the Shapiro test. In addition, I think the number should be ‘484’, please recheck it.

R.: A histogram of the p-values frequencies from the Shapiro test was added as Figure 1. To clarify the doubt with regard to the number of samples, the second paragraph of Section 4.3 now reads as follows:

“To determine the latter, it was first necessary to verify whether the Developer Scores obtained by each of the 80 MSA strategies for a given benchmark group of problem instances followed a normal distribution in order to identify whether a parametric or non-parametric test should be used. Given the two benchmarks were each divided into 3 groups, and the 80 strategies, a total of 480 samples (2 benchmarks × 3 groups × 80 strategies) were evaluated. The frequencies of the p-values obtained by a Shapiro test for each sample are shown in Fig 1. The results show that 259 of the 480 samples (≈ 53.9%) have p-values lower than 0.05 and thus, with 95% probability, these don’t follow a normal distribution. Thus, given the high probability that at least one sample does not follow a normal distribution, the non-parametric Wilcoxon signed-rank test [29] was chosen to compare the different heuristic classes.”

4. In result analysis section, I think authors could show some comparison results between the 20 variations in the first column of each table, I think the readers are also curious about the performance of NJ vs UPGMA etc.

R.: Section 4.4 was added to the paper to present two statistical comparisons, similar to that shown in Section 4.3. The first is between similarity scoring techniques and the second is between guide tree generation techniques. The penultimate paragraph of the introduction in Section 4 has also been modified to incorporate the new section.

5. HOMSTRAD is a structure-based MSA database. Each sequence in a MSA has the PDB structure in HOMSTRAD database. Thus a structurebased analysis should be added to the paper, so that the readers could know how the performances of those 80 MSA construction methods are related to the structure alignment, since protein structures could directly reflect the protein evolution. For example, instead of just calculating the DS score for G1, G2 and G3, the authors could calculate the average TM-score of the pairwise proteins in a MSA based on the MSA alignment using TMalign/TM-score tool. Then show the comparisons of the average TM-scores for those 80 MSA construction methods.

R.: This is an interesting observation. We are already aware that many consider structural alignment as the gold standard for sequence alignment, and thus have future plans on our roadmap to incorporate structure-aware techniques into our framework. However, given its current stage of development, in this paper, we have opted to only focus on trying to understand the extent to which widely adopted techniques, methods, or heuristics in existing MSA tools impact the quality of the alignments produced. In this classic alignment scenario, none of the techniques evaluated make use of information about protein structures, only the nucleotide or amino acid sequence. Thus, we thought it would be unfair to draw conclusions based on the comparison suggested. We do however believe that such a comparison would be a good starting point to evaluate enhancements to classic alignment techniques to support structural alignment. Our initial plan was to start with generating a similarity matrix based also on structural information, which will allow us to evaluate the techniques in the rest of the framework and compare the alignments obtained with TM-align/US-align. In parallel, we are working on techniques that try to predict protein structure, based on NMR-obtained atomic distance data, so that we might still be able to use the framework in situations where the structural information is not available.

But replying specifically to the review’s suggestion, given the time constraints for this review, our lack of experience in this area of structural alignments, and our unfamiliarity with the TM-score and how to provide modified structural information for the alignments produced by designed techniques for classic MSA alignments, we prefer leave this for future work so that we can take a careful and more considerate approach to make this comparison. Therefore, the following sentence was added to the conclusions in Section 5:

“The use of information regarding both the structural and functional importance of each amino acid in proteins is seen as the gold standard to evaluate alignments. In the longer term, we plan to investigate how to extend and adapt our framework for the multiple structural alignment problem and to study its applicability in comparison with structural alignment tools such as TM-align [45].”

Minor Points

1. Multiple sequence alignments actually have two meanings in recent studies. One is its traditional definition, aligning the multiple sequences with each other, inserting reasonable gaps, and letting all sequences align well. This one is mainly used for phylogeny inference, evolution analysis, etc. The other one recently used in protein/RNA structure and function prediction is more like homologous sequence detection, which aligns all homologous to a ’query’ sequence. This first one is the NP-hard problem, as the authors mentioned, while the second is mostly N times of pairwise alignments (N is the number of homologous sequences in MSA). In the “Introduction section”, please clearly mention the first one is what the authors focus on. 

R.: The following paragraph was added as the second paragraph of Introduction. “Note that this problem is computationally different, for example, from the process to find homologous sequences, which consists of obtaining multiple pairwise sequence alignments of protein or RNA sequences generally against a single reference sequence [7]. Multiple sequence alignments tend to provide more information than pairwise alignments by highlighting conserved regions of aligned residues that may be of structural and functional relevance. The focus of this paper is thus on the former, the traditional MSA problem.”

2. Recent co-evolutionary analyses are also based on MSA, like contact/distance prediction. In the “Introduction section”, please talk about this point.

R.: We have included this term and a reference in the first paragraph of the Introduction.

3. In the first paragraph of “Section 4.3”, when you first mention “20 variations”, please cite “Section 3.1 and 3.2”. In the end of “Section 3”, please clearly say how you get 20 variations by combining 5 methods in “Section 3.1” and 4 methods in “Section 3.2”. Since when I first read the word “80 combinations” and “20 variations”, I am slightly confused where those methods came from.

R.: Section 3.6 has been added to clarify the meaning of the terms "variations" and "strategies". A table has also been inserted in this Section to explain how the 20 variations were obtained and their nomenclature.

4. Text in figure 4 is overlapping.

R.: Figure 5 (the old Figure 4) has been adjusted.

---

## [Decision Letter · Decision Letter 1]

6 Jun 2023

On closing the inopportune gap with consistency transformation and iterative refinement

PONE-D-23-07871R1

Dear Dr. João Junior,

We’re pleased to inform you that your manuscript has been judged scientifically suitable for publication and will be formally accepted for publication once it meets all outstanding technical requirements.

Kind regards,

Yang Zhang

Academic Editor

PLOS ONE

Additional Editor Comments (optional):

Reviewers' comments:

Reviewer's Responses to Questions

**Comments to the Author**

1. If the authors have adequately addressed your comments raised in a previous round of review and you feel that this manuscript is now acceptable for publication, you may indicate that here to bypass the “Comments to the Author” section, enter your conflict of interest statement in the “Confidential to Editor” section, and submit your "Accept" recommendation.

Reviewer #1: All comments have been addressed

2. Is the manuscript technically sound, and do the data support the conclusions?

Reviewer #1: Yes

3. Has the statistical analysis been performed appropriately and rigorously? 

Reviewer #1: Yes

4. Have the authors made all data underlying the findings in their manuscript fully available?

Reviewer #1: Yes

5. Is the manuscript presented in an intelligible fashion and written in standard English?

Reviewer #1: Yes

6. Review Comments to the Author

Reviewer #1: I think the authors have addressed all my concerns, and I do not have any further comments. Thank you, and congratulations!

7. PLOS authors have the option to publish the peer review history of their article (what does this mean?). If published, this will include your full peer review and any attached files.

Reviewer #1: No

---

## [Editor Report · Acceptance letter]

4 Jul 2023

PONE-D-23-07871R1 

On closing the inopportune gap with consistency transformation and iterative refinement 

Dear Dr. João Jr.:

I'm pleased to inform you that your manuscript has been deemed suitable for publication in PLOS ONE. Congratulations! Your manuscript is now with our production department. 

Kind regards, 

on behalf of

Dr. Yang Zhang 

Academic Editor

PLOS ONE